# Sarcopenia Is Associated with Changes in Circulating Markers of Antioxidant/Oxidant Balance and Innate Immune Response

**DOI:** 10.3390/antiox12111992

**Published:** 2023-11-11

**Authors:** Francesco Bellanti, Aurelio Lo Buglio, Stefano Quiete, Michał Dobrakowski, Aleksandra Kasperczyk, Sławomir Kasperczyk, Gianluigi Vendemiale

**Affiliations:** 1Department of Medical and Surgical Sciences, University of Foggia, 71122 Foggia, Italy; aurelio.lobuglio@unifg.it (A.L.B.); stefano.quiete@unifg.it (S.Q.); gianluigi.vendemiale@unifg.it (G.V.); 2Department of Biochemistry, Faculty of Medical Sciences in Zabrze, Medical University of Silesia, 41-800 Katowice, Poland; michal.dobrakowski@poczta.fm (M.D.); olakasp@poczta.onet.pl (A.K.); skasperczyk@sum.edu.pl (S.K.)

**Keywords:** redox balance, innate immunity, sarcopenia

## Abstract

(1) Background: The involvement of redox balance alterations and innate immunity is suggested to play a key role in the pathogenesis of sarcopenia. This investigation aimed to define and relate modifications in circulating markers of redox homeostasis and the innate immune response in human sarcopenia. (2) Methods: A total of 32 subjects aged >65 years old and affected by sarcopenia according to the second “European Working Group on sarcopenia in older people” guidelines were compared with 40 non-sarcopenic age-matched controls. To assess systemic redox homeostasis, reduced (GSH) and oxidized (GSSG) blood glutathione and plasma malondialdehyde (MDA)– and 4-hydroxy-2,3-nonenal (HNE)–protein adducts were measured. Immune cells and circulating interleukins were determined to compare the innate immune response between both groups. (3) Results: Impaired redox balance in sarcopenic patients, characterized by a high blood GSSG/GSH ratio and plasma MDA/HNE–protein adducts, was sustained by reduced antioxidants in peripheral blood mononuclear cells. Furthermore, sarcopenic patients showed higher neutrophil-to-lymphocyte ratios and interleukin (IL)-4, IL-6, IL-10, and tumor necrosis factor (TNF) with respect to non-sarcopenic patients. Linear regression analysis resulted in a strong association between redox balance and immune response markers in the sarcopenic group. (4) Conclusions: These results support the interplay between redox homeostasis alteration and disruption of the innate immune response in the pathogenesis of sarcopenia.

## 1. Introduction

Population age is dramatically rising all over the world, with a consequent increase in people with lower physical function and altered quality of life [1]. Sarcopenia is a common age-associated condition, defined as a reduction in both skeletal muscle mass and function [2]. In addition to causing disability, sarcopenia is a determinant risk factor for falls, fractures, hospitalizations, and death [3].

The pathogenesis of sarcopenia is complex and multifactorial, but the identification of key determinant mechanisms is essential to identify reliable markers and to provide effective therapies. A decrease in muscle mass and strength may be sustained by chronic low-grade inflammation that has been reported in aged life, defined as “inflamm-aging” [4,5]. This condition is characterized by increased levels of pro-inflammatory cytokines (such as interleukin 6, IL-6, and tumor necrosis factor (TNF)) and reduced amounts of anti-inflammatory mediators (including IL-4 and IL-10) [6]. Indeed, previous studies reported increased levels of IL-6 and TNF, but also higher IL-4 and IL-10, in patients with sarcopenia with respect to healthy individuals [7,8]. Since the cytokine network finely tunes both the immune response and the inflammatory process, characterization of the mechanisms that regulate immunity in sarcopenia is of utmost importance.

A further pathogenic mechanism of age-associated diseases—including sarcopenia—relies on the dysregulation of redox homeostasis [9,10]. This is defined as a subtle balance between oxidants—or reactive species—and antioxidants, i.e., reducing compounds [11]. The production of reactive species by skeletal muscle cells during aging may overwhelm antioxidants, with consequent oxidative stress and tissue damage [12]. In a previous study, we reported that circulating markers of oxidative stress are increased in sarcopenic patients, suggesting that redox balance alteration contributes to the pathogenesis of sarcopenia [13]. Altered redox balance is also reported in skeletal muscle denervation, even though the role of denervation-dependent oxidative stress on muscle atrophy is controversial [14]. Reactive species and antioxidant compounds are also able to modulate the immune response [15]. In particular, the metabolic reprogramming of immune cells, macrophage and T cell polarization, cytokine production, immune tolerance, pathogen sensing, chemotaxis, and phagocytosis are all modulated by redox-dependent mechanisms [16]. Even though the interplay between redox balance alterations and the immune response was suggested as a determinant for sarcopenia by several authors [4,17,18], clinical evidence of this potential relationship is lacking.

Expanding upon our previous observation, this study aimed to evaluate alterations in circulating biomarkers of both the immune response and redox balance in human sarcopenia, to establish a possible cross-sectional association between oxidative stress and immunosenescence in the pathogenesis of this age-related condition.

## 2. Materials and Methods

### 2.1. Study Population and Design

This study was designed as a cross-sectional, observational, single-center investigation, based on individuals aged 65 years or older who could independently be evaluated for walking speed, grip strength, and muscle mass, consecutively recruited from January 2022 to April 2023 at the Department of “Medicina Interna e dell’Invecchiamento”, “Policlinico Riuniti”, in Foggia (Italy). The exclusion criteria were as follows: (1) an age of <65 years old; (2) unwillingness to participate in the study; (3) an inability to fulfill the required inspection items independently; (4) active inflammatory disease within the last 3 months; (5) malignant tumors; (6) immune system disease; (7) active liver disease; (8) acute cardiovascular and cerebrovascular disease; (9) use of drugs affecting body composition or redox balance; (10) daily alcohol intake >40 g.

Participants were divided into two groups (not sarcopenic and sarcopenic) based on the diagnosis of sarcopenia, defined according to the criteria of the European Working Group on Sarcopenia in Older People 2 (EWGSOP2), as described previously [2]. Demographic characteristic (age, gender), lifestyle (smoking, drinking, and physical activity according to self-report activity logs), chronic diseases (diabetes mellitus, hyperlipidemia, hypertension, chronic liver disease, and chronic kidney disease), pharmacotherapy, and anthropometric data were collected. All patients underwent geriatric multidimensional evaluation, which included activities of daily living (ADL) and instrumental activities of daily living (IADL), the mini nutritional assessment (MNA), the geriatric depression scale—short form, and the mini mental state examination. The study protocol was approved by the Institutional Ethics Committee of the “Policlinico Riuniti” in Foggia (n. 3263/2021) and performed according to the Declaration of Helsinki. Data of interest for the analysis were anonymously collected in a dedicated database. All participants gave written informed consent.

### 2.2. Assessment of Sarcopenia

Muscle strength was assessed by performing a handgrip strength test, using a Kern MAP 80K1S Hand Grip Dynamometer (Kern, Balingen, Germany). Subjects were instructed to apply as much handgrip pressure as possible; measurements were repeated three times, and the highest score (kilograms) was recorded. Low grip strength was defined as <27 kg in men and <16 kg in women [19].

Muscle quantity was assessed via bioelectrical impedance using a BIA 101-F device (Akern/RJL, Florence, Italy). The BIA measure was performed via a standard technique using a single frequency of 50 KHz and the placement of four electrodes in a distal position. The participant was measured while in a standing position. The values of reactance and resistance were then recorded once the patient was stabilized. Low muscle quantity was characterized by an appendicular skeletal mass (ASM)/height^2^ < 7.0 kg/m^2^ in men and <5.5 kg/m^2^ in women [20].

Muscle performance was assessed by measuring gait speed on a 4 m course (preceded by 1 m acceleration course and followed by 1 m deceleration course), on a marked floor, after instructing patients about the different zones and recommending that they walk at their normal pace. Each test was repeated twice, and low performance was defined as a speed ≤ 0.8 m/s [21].

Diagnosis of sarcopenia was performed in subjects showing low muscle strength and quantity, while severe sarcopenia was diagnosed by further reporting low muscle performance [2].

### 2.3. Laboratory Measurements

Blood samples were obtained after an overnight fast from a brachial vein, between 8 a.m. and 9 a.m., and immediately processed. Standard laboratory measurements included a hemocromocytometric test, and tests for serum glucose, glycated hemoglobin, insulin, total cholesterol, low-density lipoprotein (LDL) and high-density lipoprotein (HDL)-cholesterol, triglycerides, creatinine, albumin, uric acid, 25-(OH)-vitamin D, protein electrophoresis, erythrocyte sedimentation rate (ESR), C-reactive protein (CRP), fibrinogen, and ferritin. HOMA-IR index was calculated according to the following formula: HOMA-IR = [glucose] (mg/dL) × [insulin] (µU/mL)/405.

Oxidized (GSH) and reduced (GSSG) glutathione were determined in whole blood, as previously described [22].

Plasma fluorescent adducts formed between peroxidation-derived aldehydes (HNE and MDA) and proteins were measured via spectrofluorimetry, as previously reported [23].

The concentrations of serum cytokines and growth factors, including IL-1α, IL-1β, IL-2, IL-4, IL-6, IL-8, IL-10, tumor necrosis factor (TNF), interferon (IFN)-γ, monocyte chemoattractant protein-1 (MCP-1), epidermal growth factor (EGF), and vascular endothelial growth factor (VEGF), were measured using an EV 3513 cytokine biochip array and competitive chemiluminescence immunoassays (Randox Laboratories Ltd., Crumlin, UK), according to the manufacturer’s instructions, using the Randox Evidence Investigator [24].

### 2.4. RNA Isolation and Quantitative Real-Time Reverse-Transcription Polymerase Chain Reaction (qRT-PCR)

Total cellular RNA was isolated from peripheral blood mononuclear cell (PBMC) samples using an RNeasy Kit (Qiagen, Hilden, Germany) according to the manufacturer’s instructions. To ensure minimum in vitro impact on the activation status of cells, we employed modified gradient separation. PBMCs were immediately isolated via rapid Ficoll-Histopaque centrifugation for 30 min at 900× *g*. Total cellular RNA was extracted using the RNeasy kit and immediately stored at −80 °C. Samples were quantified via absorption spectrophotometry, and RNA integrity was confirmed using nondenaturing agarose gel electrophoresis. cDNA was obtained using a random hexamer primer and a SuperScript III Reverse Transcriptase kit as described by the manufacturer (Invitrogen, Frederick, MD, USA). A PCR mastermix containing the specific primers superoxide dismutase 1 (SOD1) (forward, TGT GGG GAA GCA TTA AAG G; reverse, CCG TGT TTT CTG GAT AGA GG); catalase (CAT) (forward, GCC ATT GCC ACA GGA AAG TA; reverse, CCA ACT GGG ATG AGA GGG TA); glutathione reductase (GR) (forward, GGA GAC CTC ACC CTG TAC C; reverse, GTC ATT CAC CAT GTC CAC C); glutathione synthetase (GS) (forward, ACC TCC ACC GTA TAT TTG AG; reverse, TTG CCC CAG ACA GCC ATC TT); and glyceraldehyde-3-phosphate dehydrogenase (GAPDH) (forward, CAA GGC TGA AAC GGG AA; reverse: GCA TCG CCC CAC TTG ATT TT) was added, along with AmpliTaq Gold DNA polymerase (Applied Biosystems, Foster City, CA, USA). Real-time quantification of mRNA was performed via a SYBRGreen I assay and evaluated using an iCycler detection system (Bio-Rad Laboratories). The threshold cycle (CT) was determined, and the relative gene expression was subsequently calculated as follows: fold change = 2^−Δ(ΔCT)^, where ΔCT = CT_target_ − CT_housekeeping_ and Δ(ΔCT) = ΔCT_treated_ − ΔCT_control_.

### 2.5. Statistical Analysis

Data are expressed as counts and percentages for qualitative values, and as mean ± standard deviation of the mean (SDM) for quantitative variables. Gaussian distribution of the samples was evaluated using a Kolmogorov–Smirnov test.

The significance of differences between 2 groups (sarcopenic vs. non-sarcopenic) was assessed via Student’s *t*-test (continuous variables) or in contingency tables using Pearson’s Chi squared test and Fisher’s exact test (categorical variables). A sample size of 28 subjects in each group was calculated to detect the difference of 10.35 in GSSG/GSH according to our previous work [13] at 90% power and a 95% level of significance. The significance of differences between more than 2 groups was assessed via one-way analysis of variance (ANOVA) after ascertaining normality using the Kolmogorov–Smirnov test; the Tukey–Kramer test was applied as a post hoc test. When the data were not normally distributed, non-parametric tests (Mann–Whitney (M-W, comparison of two groups) and Kruskal–Wallis with Dunn’s method (comparison of multiple groups)) were employed.

Linear regression models were used to analyze the association between circulating markers of redox balance alteration with markers of inflammation/immune response. Pearson’s (normally distributed data variables) and Spearman’s (non-normally distributed data variables) correlation analyses were performed to determine the strength of the linear relationship between different parameters. A heatmap was plotted using https://www.bioinformatics.com.cn/en (accessed on 23 October 2023), a free online platform for data analysis and visualization. Finally, a series of crude linear regression models was run to predict the association between variables of muscle function/mass/performance and immune response markers with GSSG/GSH as redox balance markers. From the regression models, point estimates with 95% confidence intervals (CI) were computed in order to quantify the amount of change in GSSG/GSH per unit change in the predictor variables, and the statistical significance of predictor variables was assessed via F-tests. R^2^ values to assess the proportion of GSSG/GSH variance explained by the predictor variables were also computed. In all models, handgrip strength, ASM/height^2^, gait speed, neutrophil count, NLR, IL-1α, IL-6, IL-10, VEGF, and MCP-1 were treated as continuous, linear predictor variables.

All tests were two-sided, and significance was accepted at *p* < 0.05.

Statistical analysis was performed using SPSS version 23 (SPSS, Inc., Chicago, IL, USA) and the package Graph-Pad Prism 6 for Windows (GraphPad Software, Inc., San Diego, CA, USA).

## 3. Results

### 3.1. Baseline Characteristics

During the observation period, 72 individuals were recruited; of these, 32 (44.4%) were diagnosed with sarcopenia according to the EWGSOP2 criteria, while 40 (55.6%) subjects were included in the “No sarcopenia” group. The baseline characteristics of the recruited individuals, stratified according to the diagnosis of sarcopenia, are shown in Table 1.

Both groups were similar in terms of age, sex, smoking, alcohol use, physical activity, co-morbidity, and polypharmacotherapy; however, despite comparable weight, sarcopenic patients showed lower BMI and waist circumference than individuals with no sarcopenia. Furthermore, reduced levels of hemoglobin, HDL-cholesterol, and 25-(OH)-vitamin D, and higher insulin levels, HOMA-IR indexes, and creatinine levels, were reported in sarcopenic compared to non-sarcopenic subjects.

Differences in measures related to muscle strength, quantity, and performance, as well as results related to multidimensional evaluation, are reported in Table 2. Sarcopenic patients exhibited lower handgrip strength, ASM/height^2^, and gait speed with respect to people with no sarcopenia. Of note, scores from all the tests included in our geriatric multidimensional evaluation protocol were reduced in the sarcopenia compared to the non-sarcopenia group (Table 2).

### 3.2. Circulating Markers of Redox Balance and Immunity Are Altered in Sarcopenia

Redox homeostasis in all study participants was analyzed by measuring circulating glutathione balance and proteins oxidatively modified through lipoperoxidative reactions. Table 3 reports data related to circulating markers of redox homeostasis.

Of note, a significant reduction in reduced glutathione (GSH) was observed in sarcopenic patients compared with subjects with no sarcopenia, while oxidized glutathione (GSSG) levels were similar in both groups. Consequently, the GSSG/GSH ratio was higher in the sarcopenia than in the no-sarcopenia group. Furthermore, a significant increase in aldehyde–protein adducts was observed in patients with sarcopenia with respect to non-sarcopenic individuals. Taken together, these results suggest that sarcopenia is characterized by high circulating markers of oxidative stress, which is mostly sustained by impaired GSH synthesis/recycling.

To verify this hypothesis, the analysis of genes encoding for enzymes related to GSH synthesis/recycling (glutathione reductase (GR) and glutathione synthetase (GS); a scheme showing how these enzymes contribute to GSH metabolism is represented in Appendix A), as well as other antioxidant enzymes (superoxide dismutase 1 (SOD1) and catalase (CAT)), was performed via real-time RT-PCR on mRNA extracted from peripheral blood mononuclear cells (PBMCs) of both groups. As reported in Figure 1, the expression of SOD1, CAT, GR, and GS was significantly reduced in patients with sarcopenia compared to non-sarcopenic individuals.

To evaluate inflammation and the immune response in the study participants, systemic markers were measured, and the results are reported in Table 4. 

Of interest, all markers were increased in sarcopenic compared to non-sarcopenic individuals. Even though WBC and lymphocyte counts was similar between groups, higher neutrophils and NLR were reported in patients with sarcopenia compared to those with no sarcopenia.

To complete the study of immunity, the serum levels of 12 cytokines and growth factors were evaluated in the study participants, and the results are shown in Table 5.

No differences were reported for IL-1α, IL-1β, IL-2, IL-8, IL-10, IFN-γ, and EGF between the two groups. Of note, serum IL-4 levels were reduced, while IL-6, TNF, VEGF, and MCP-1 concentrations were increased, in patients with sarcopenia compared to those with no sarcopenia. Taken together, these results sustain a circulating pro-inflammatory status in sarcopenic compared to non-sarcopenic subjects.

### 3.3. Circulating Markers of Redox Homeostasis Are Associated with Innate Immune Response in Sarcopenia

To investigate the potential relationship between markers of redox balance and inflammation/immunity, a correlation matrix was generated, as graphically represented in Figure 2.

Serum insulin levels were not related to muscle strength/mass/performance, redox balance, or immunity markers. Interestingly, the indexes of muscle function/mass/performance were positively related to circulating markers of redox balance and inflammation/immunity.

Of note, strong positive correlations were observed between all the studied markers of redox balance and neutrophil count, as well as NLR. Furthermore, such markers were positively correlated with several pro-inflammatory cytokines, including IL-1α, IL-6, TNF, VEFG, and MCP-1, and negatively correlated with the anti-inflammatory cytokine IL-10.

The relationships observed were further confirmed via linear regression analysis. In particular, strong (* r*^2^ > 0.5) positive associations between GSSG/GSH and NLR (*r*^2^ = 0.612, *p* < 0.001), TNF (* r*^2^ = 0.760, *p* < 0.001), and VEGF (* r*^2^ = 0.576, *p* < 0.001); between HNE–protein adducts and NLR (* r*^2^ = 0.718, *p* < 0.001), TNF (* r*^2^ = 0.767, *p* < 0.001), and VEGF (* r*^2^ = 0.618, *p* < 0.001); and between MDA–protein adducts and NLR (* r*^2^ = 0.696, *p* < 0.001), TNF (* r*^2^ = 0.709, *p* < 0.001), VEGF (* r*^2^ = 0.563, *p* < 0.001), and MCP-1 * r*^2^ = 0.510, *p* < 0.001 were reported (Appendix A).

Table 6 shows the results of unadjusted linear regression models assessing the ability of muscle strength/mass/performance markers and serum immune response factors to predict GSSG/GSH values, chosen as a circulating marker of redox balance. Among all the considered markers, handgrip strength, NLR, and IL-6 showed significant positive associations with GSSG/GSH values. In particular, GSSG/GSH values increased by 2.3, by 4.8 and by 2.9 for each SD increase in handgrip strength, NLR, and IL-6, respectively. The *R*^2^ of these models were 0.30, 0.42, and 0.15, indicating that handgrip strength, NLR, and IL-6 explained, respectively, 30%, 42%, and 15% of the variance in GSSG/GSH.

## 4. Discussion

To our knowledge, this is the first investigation demonstrating that alterations in circulating markers of redox balance are associated with modifications to the immune response in human sarcopenia.

In a biological milieu, imbalance between oxidants and antioxidants in favor of the former is defined as oxidative stress [25]. This condition has been linked to loss of muscle fibers, a decrease in muscle quality, and the impairment of muscle performance [26,27]. Furthermore, the overproduction of reactive species is triggered by denervation—caused by the age-related degeneration of motor neurons and neuromuscular junctions—which contributes to the pathogenesis of sarcopenia [28,29]. Indeed, we have already described increased circulating markers of oxidative stress in sarcopenic patients, as demonstrated by high levels of GSSG/GSH and HNE– and MDA–protein adducts, but we did not elucidate the potential underlying mechanisms and did not explore possible associations with other pathogenic factors [13]. The results from the present study confirm our previous observation and strengthen our understanding of the role of redox homeostasis in the maintenance of muscle structure and function. In particular, GSSG/GSH redox potential is considered a standard electrochemical factor representing a common, powerful regulator of redox reactions modulating biological events [30,31]. As formerly reported, the elevated GSSG/GSH levels in sarcopenia are sustained by decreased GSH rather than raised GSSG, suggesting that the progression of oxidative injury is supported by defective GSH synthesis/recycling. GSH is a tripeptide composed of cysteine, glutamic acid, and glycine, with strong antioxidant properties; it is synthetized by γ-glutamylcysteine synthetase and glutathione synthetase (GS) [32]. To preserve redox homeostasis, GSH scavenges reactive species and is oxidized to GSSG, so the maintenance of adequate supplies of GSH is provided by glutathione reductase (GR) [33]. During aging, glutathione content decreases in the skeletal muscle tissue of rodents [34]. Furthermore, skeletal muscle disuse leads to perturbations of circulating GSH and GSSG/GSH in humans with hip fracture [35]. We investigated the mechanisms accounting for reduced GSH and increased GSSG/GSH by evaluating the expression of antioxidant genes, including GS and GR, in the PBMCs of recruited individuals. Unreliable GSH synthesis/recycling was then supported by reduced GS and GR expression in PBMCs from sarcopenic patients. Furthermore, our data showed decreased expression of SOD1 and CAT, which encode for the two main endogenous antioxidant enzymes. Overall, these reports are consistent with reduced systemic antioxidant defense as a possible cause of redox disbalance in sarcopenia. Indeed, similar to the brain, skeletal muscle is the greatest oxygen utilizer in the human body, constantly producing reactive species both at rest and during contraction [36]. Thus, redox homeostasis in skeletal muscle is preserved through a consistent antioxidant system [37]. It is then conceivable that the deterioration of antioxidant capacity could trigger oxidative damage in sarcopenia. This hypothesis is sustained by previous observations on pre-clinical models, in which SOD1 depletion causes early sarcopenia, or CAT overexpression reduces the age-dependent loss of muscle quantity and performance [38,39]. Nevertheless, our data need to be interpreted with caution, since no direct measurement of redox balance was performed in skeletal muscle.

Recent research has been focused on the role of inflammation and the immune response in the development of sarcopenia. Persistent low-grade inflammation which characterizes aging, defined as “inflammaging” and caused by disruption of the immune system, may negatively influence skeletal muscle homeostasis, contributing to the loss of muscle mass and strength [5]. Age-related immune alterations typical of sarcopenia are characterized by increased circulating levels of pro-inflammatory cytokines and reduced anti-inflammatory mediators [40,41]. Our data support a chronic low-grade pro-inflammatory systemic status in sarcopenia, as suggested by higher circulating levels of ESR, CRP, fibrinogen, ferritin, and α2-globulins, which are markers of inflammation. This observation is further sustained by high circulating levels of IL-6, TNF, VEGF, and MCP-1, and reduced concentrations of IL-4, which foster a pro-inflammatory status. Our results are in line with previous studies that observed elevated levels of circulating pro-inflammatory markers in sarcopenic patients [7,8,42,43].

Inflammaging is associated with immunosenescence, which is characterized by a decline in both the innate and adaptive immune responses [44]. As a consequence of immunosenescence, impaired cell–cell interactions and altered signaling pathways vehiculate disrupted signals to the skeletal muscle [45]. Innate immunity mainly involves neutrophils, macrophages, natural killer cells, and dendritic cells (DCs), while adaptive immunity involves antigen/specific T/B lymphocytes [46]. In particular, inflammaging appears to be sustained by innate immune cell dysfunction, thus contributing to sarcopenia [47,48]. Data from this study show a relative increase in circulating neutrophils and NLR in sarcopenia. NLR can be considered a biomarker that couples both arms of the immune response, since neutrophils mostly support innate immunity, while adaptive immunity is mainly dependent on lymphocytes [49]. Considering that IL-6 and TNF are included among the most important mediators released by innate immune cells [50], our data provide evidence of activated innate immunity in sarcopenic patients. This evidence is further supported by increased circulating levels of MCP-1 and VEGF, which play critical roles in innate immunity [51,52].

Altered redox balance is associated with dysregulation of the immune–inflammatory response [16]. The interaction between reactive species and innate immunity effectors is orchestrated by several redox-dependent transcription factors, such as nuclear factor-κB (NF-κB) [53]. For the first time, we report a strong relationship between circulating markers of redox homeostasis and serum cytokines in human sarcopenia, as well as biomarkers of innate immunity (neutrophil count and NLR). Even though we could not establish a causal relationship between redox homeostasis alteration and disrupted immune response, this association supports the hypothesis of crosstalk between redox balance and the immune system in the pathogenesis of this age-related condition [17]. Going beyond the first study from our team, our observation in samples from sarcopenic patients suggests that oxidative stress is the result of impaired antioxidant systems and contributes to alterations in the innate immune response. On the other hand, it is also conceivable that low-grade inflammation could contribute to the overproduction of reactive species, initiating an auto-toxic loop that facilitates a pathophysiologic pathway in sarcopenia. However, refined mechanistic studies are needed to define whether reactive species trigger immunosenescence and inflammaging, causing disruption to the innate immune response, or whether age-related altered innate immunity leads to redox disbalance in sarcopenia.

This study displays several limitations. First of all, it is conceivable that the small sample size could not enable us to draw reliable conclusions. Secondly, this investigation was conducted in a single center, so it may have presented some bias. We registered a higher frequency of sarcopenia compared to previous reports [54]; this is mostly due to the study setting (recruitment in a hospital department), and this aspect should be considered when interpreting the results. Nonetheless, even though future multicentric studies are needed to confirm our findings, our observations may be considered an important step in understanding the association between redox balance alterations and the disruption of innate immunity in sarcopenia. Even though we could not define the initiating role of oxidative stress or immunosenescence as precipitant or progressive factors in sarcopenia, speculations can be suggested. The impairment of redox homeostasis and the immune response are both components of multisystem efficiency reduction that characterize aging and age-related diseases. In addition, several chronic diseases may both reduce antioxidant defense and alter immune status. Since most aged individuals present with co-morbidities, redox balance is often shifted toward the oxidative state, and the immune response frequently activates a pro-inflammatory status, increasing susceptibility to sarcopenia.

In conclusion, our study supports the interplay between redox homeostasis alteration and disruption of the innate immune response in the pathogenesis of sarcopenia. Since modifications of redox balance in sarcopenia mostly rely on impaired antioxidant status, it is conceivable that supplementations could exert a beneficial impact on the immune response and inflammation, reducing skeletal muscle injury. Future pre-clinical and clinical studies with suitable methodological approaches are encouraged to define and test specific compounds, as well as timing and dosage, for the prevention and treatment of sarcopenia.

## Figures and Tables

**Figure 1 antioxidants-12-01992-f001:**
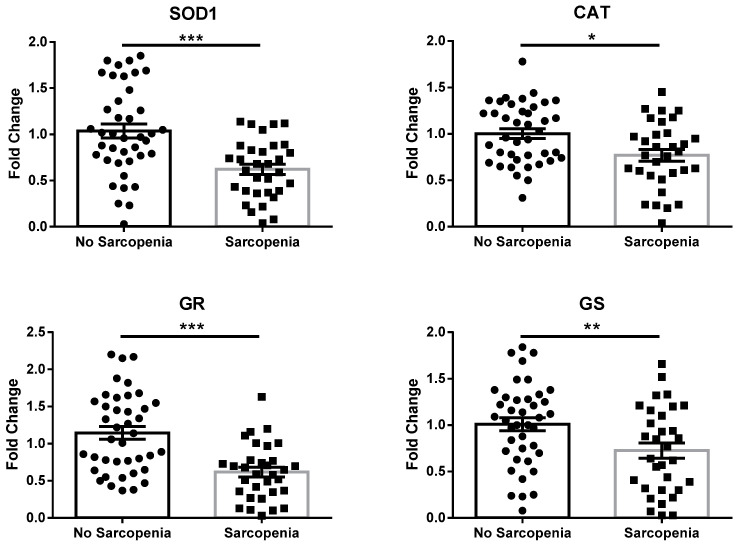
mRNA expression of antioxidant enzymes in PBMCs extracted from subjects included in this study, according to absence or presence of sarcopenia. Abbreviations: SOD1, superoxide dismutase 1; CAT, catalase; GR, glutathione reductase; GS, glutathione synthetase. Data are represented as mean ± standard error. Statistical analysis was performed via Student’s *t*-test. * = *p* < 0.05; ** = *p* < 0.01; *** = *p* < 0.001.

**Figure 2 antioxidants-12-01992-f002:**
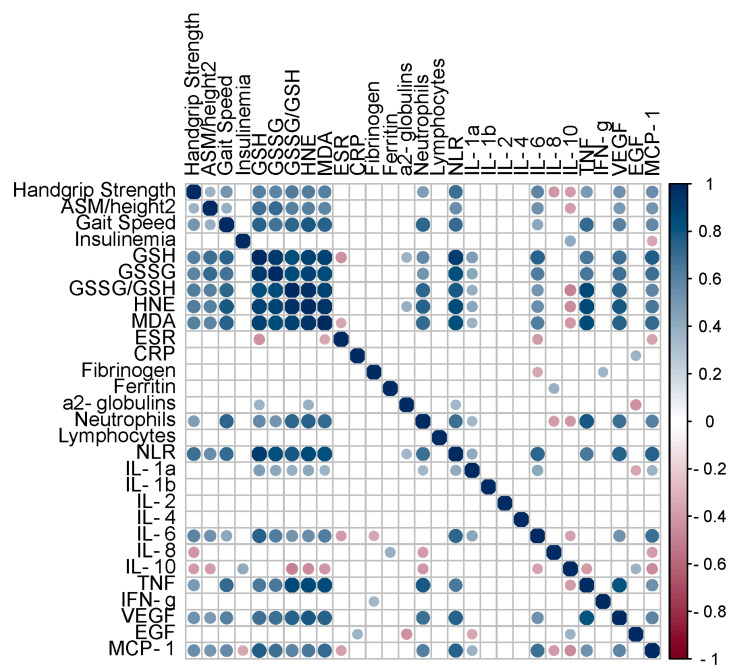
Pearson’s correlation matrix for indexes of muscle function/mass/performance, insulinemia, circulating markers of redox balance, and markers of systemic inflammation/immune response. Only significant (*p* value < 0.05) correlations are represented with full circles. Positive correlations (0 < *r* < 1.0) are displayed in blue color, and negative correlations (−1.0 < *r* < 0) are displayed in red color. Color intensity is proportional to the correlation coefficients. Abbreviations: GSH, reduced glutathione; GSSG, oxidized glutathione; HNE, hydroxynonenal–protein adducts; MDA, malondialdehyde–protein adducts; ESR, erythrocyte sedimentation rate; CRP, C-reactive protein; NLR, neutrophil-to-lymphocyte ratio; IL, interleukin; TNF, tumor necrosis factor; IFN, interferon; VEGF, vascular endothelial growth factor; EGF, epidermal growth factor; MCP-1, monocyte chemoattractant protein-1.

**Table 1 antioxidants-12-01992-t001:** Baseline characteristics of subjects included in this study, according to absence or presence of sarcopenia.

Variable	No Sarcopenia *n* = 40 (55.6%)	Sarcopenia *n* = 32 (44.4%)	*p* Value
Age, years	74.8 (±5.1)	76.4 (±4.5)	0.168
F Sex, *n* (%)	14 (45.0%)	10 (31.2%)	0.737
Smoking, *n* (%)	10 (25.0%)	12 (37.5%)	0.252
Alcohol consumer, *n* (%)	21 (52.5%)	17 (53.1%)	0.958
Low physical activity, *n* (%)	8 (20%)	9 (28.1%)	0.420
Diabetes mellitus, *n* (%)	22 (55.0%)	19 (59.3%)	0.709
Hyperlipidemia, *n* (%)	20 (50%)	20 (62.5%)	0.289
Hypertension, *n* (%)	28 (70.0%)	20 (62.5%)	0.502
Chronic liver disease, *n* (%)	1 (2.5%)	2 (6.2%)	0.429
Chronic kidney disease, *n* (%)	5 (12.5%)	8 (25%)	0.171
Co-morbidities > 3, *n* (%)	6 (15%)	9 (28.1%)	0.173
Pharmacotherapy > 6, *n* (%)	6 (15%)	10 (31.2%)	0.099
Weight, kg	70.3 (±7.4)	71.9 (±6.4)	0.456
BMI, kg/m^2^	28.6 (±2.8)	26.1 (±1.9)	**0.024**
Waist circumference, cm	108.2 (±15.4)	100.4 (±11.9)	**0.021**
Tricipital fold, cm	20.4 (±9.6)	18.2 (±7.9)	0.300
Hemoglobin, g/dL	12.4 (±2.1)	9.5 (±2.2)	**<0.001**
Glucose, mg/dL	109.4 [72.0, 134.2]	115.7 [75.7, 143.1]	0.707
HbA1c, %	6.67 [5.97, 8.40]	6.16 [5.52, 7.84]	0.358
Insulin, μUI/mL	6.1 [1.1, 12.8]	18.0 [4.1, 36.7]	**<0.001**
HOMA-IR index	1.78 (±0.18)	5.44 (±0.87)	**<0.001**
Total cholesterol, mg/dL	157.7 [110.6, 203.8]	132.4 [100.7, 177.5]	**0.013**
LDL-cholesterol, mg/dL	99.2 [56.8, 135.0]	82.2 [52.8, 117.2]	0.063
HDL-cholesterol, mg/dL	49.3 [26.0, 72.1]	37.1 [19.6, 59.5]	**0.020**
Triglycerides, mg/dL	107.6 (±43.0)	96.9 (±60.1)	0.408
Creatinine, mg/dL	1.29 [0.44, 1.67]	1.68 [1.26, 2.31]	**0.020**
Albumin, g/dL	3.12 (±0.62)	2.8 (±0.5)	0.450
25-(OH)-vitamin D, ng/mL	29.1 [10.1, 51.8]	15.1 [5.5, 39.8]	**0.012**

Data are expressed as mean (± standard deviation), median [interquartile range], or *n* (percentage) as appropriate. Abbreviations: F, female; BMI, body mass index; LDL, low-density lipoprotein; HDL, high-density lipoprotein. *p* Values < 0.05 are considered statistically significant (in bold).

**Table 2 antioxidants-12-01992-t002:** Characteristics related to muscle strength, quality, and performance, as well as multidimensional assessment, in subjects included in this study, according to absence or presence of sarcopenia.

Variable	No Sarcopenia *n* = 40 (55.6%)	Sarcopenia *n* = 32 (44.4%)	*p* Value
Handgrip strength, kg	26.8 ± 9.1	16.2 ± 9.2	**<0.001**
ASM/height^2^, kg/m^2^	9.4 ± 2.3	5.2 ± 1.9	**<0.001**
Gait speed, m/s	0.9 [0.4, 1.5]	0.4 [0.1, 0.9]	**<0.001**
ADL, score	6 [5, 6]	4 [1, 6]	**<0.001**
IADL, score	5 [4, 8]	2 [1, 5]	**<0.001**
MNA, score	22.6 ± 6.2	17.9 ± 5.9	**0.002**
GDS-SF, score	3 [1, 6]	5 [3, 7]	**0.001**
MMSE, score	24.1 ± 4.9	16.7 ± 8.4	**<0.001**

Data are expressed as mean (± standard deviation) or median [interquartile range] as appropriate. Abbreviations: ASM, appendicular skeletal muscle mass; ADL, activities of daily living; IADL, instrumental activities of daily living; MNA, mini nutritional assessment; GDS-SF, geriatric depression scale—short form; MMSE, mini mental state examination. *p* Value < 0.05 are considered statistically significant (in bold).

**Table 3 antioxidants-12-01992-t003:** Changes in oxidized (GSH) and reduced (GSSG) blood glutathione levels and GSSG/GSH ratio, as well as serum levels of fluorescent hydroxynonenal (HNE)—and malondialdehyde (MDA, panel E)–protein adducts, in subjects included in this study, according to the absence or presence of sarcopenia.

Variable	No Sarcopenia *n* = 40 (55.6%)	Sarcopenia *n* = 32 (44.4%)	*p* Value
GSH, μM	60.6 (±25.1)	36.4 (±24.5)	**<0.001**
GSSG, μM	5.14 (±3.21)	6.43 (±3.17)	0.093
GSSG/GSH, %	8.42 (±4.75)	17.7 (±10.6)	**<0.001**
HNE–protein adducts (AUF)	25.4 (±19.8)	49.6 (±28.8)	**<0.001**
MDA–protein adducts (AUF)	58.9 (±45.4)	89.9 (±53.2)	**0.009**

Data are expressed as mean (± standard deviation). AUF, arbitrary units of fluorescence. *p* Value < 0.05 are considered statistically significant (in bold).

**Table 4 antioxidants-12-01992-t004:** Comparison of circulating markers of inflammation and immune response in subjects included in this study, according to absence or presence of sarcopenia.

Variable	No Sarcopenia *n* = 40 (55.6%)	Sarcopenia *n* = 32 (44.4%)	*p* Value
ESR, mm/h	10.4 [4.3, 21.2]	23.8 [2.1, 89.4]	**0.032**
CRP, mg/L	4.98 [1.12, 12.6]	21.5 [3.46, 64.8]	**0.019**
Fibrinogen, mg/dL	208.4 (±184.5)	417.3 (±224.0)	**<0.001**
Ferritin, ng/mL	314.6 (±118.3)	524.1 (±334.7)	**<0.001**
α2-globulins, g/dL	5.46 (±1.89)	9.64 (±6.82)	**<0.001**
γ-globulins, g/dL	0.72 (±0.55)	0.91 (±0.80)	0.237
WBC, n/mm^3^	6762 [4412, 14,131]	8291 [4016, 15,322]	0.383
Neutrophils, n/mm^3^	3814 [1964, 7852]	5829 [2366, 9983]	**0.025**
Lymphocytes, n/mm^3^	1098 (612, 3056]	1791 [534, 3827]	0.057
Monocytes, n/mm^3^	0.77 [0.12–1.34]	0.53 [0.08–1.21]	0.129
NLR	2.19 (±1.94)	3.25 (±2.52)	**0.047**

Data are expressed as mean (± standard deviation) or median [interquartile range]. Abbreviations: ESR, erythrocyte sedimentation rate; CRP, C-reactive protein; WBC, white blood cells; n, number; NLR, neutrophil-to-lymphocyte ratio. *p* Values < 0.05 are considered statistically significant (in bold).

**Table 5 antioxidants-12-01992-t005:** Comparison of circulating cytokines and growth factors in subjects included in this study, according to absence or presence of sarcopenia.

	No Sarcopenia *n* = 40 (55.6%)	Sarcopenia *n* = 32 (44.4%)	*p* Value
IL-1α, pg/mL	0.96 [0.50, 1.95]	0.73 [0.50, 1.58]	0.206
IL-1β, pg/mL	0.95 [0.44, 1.96]	0.82 [0.46, 1.84]	0.503
IL-2, pg/mL	1.15 (±0.65)	1.21 (±0.61)	0.630
IL-4, pg/mL	1.34 [0.10, 2.64]	0.10 [0.00, 0.10]	**0.002**
IL-6, pg/mL	15.6 (±12.9)	25.7 (±26.2)	**0.036**
IL-8, pg/mL	529 [128, 1457]	269 [46, 804]	0.101
IL-10, pg/mL	1.00 (±0.82)	1.41 (±1.17)	0.085
TNF, pg/mL	3.31 (±2.07)	4.53 (±1.59)	**0.008**
IFN-γ, pg/mL	0.28 (±0.33)	0.35 (±0.39)	0.555
VEGF, pg/mL	292 (±115)	417 (±182)	**<0.001**
EGF, pg/mL	10.0 (±14.4)	14.9 (±13.8)	0.148
MCP-1, pg/mL	306 (±129)	414 (±163)	**0.002**

Data are expressed as mean (± standard deviation) or median [interquartile range]. Abbreviations: IL, interleukin; TNF, tumor necrosis factor; IFN, interferon; VEGF, vascular endothelial growth factor; EGF, epidermal growth factor; MCP, monocyte chemoattractant protein. *p* Values < 0.05 are considered statistically significant (in bold).

**Table 6 antioxidants-12-01992-t006:** Unadjusted linear regression predictors of GSSG/GSH among subjects observed in this study.

Predictor (per SD Increase)	Change (95% C.I.) in GSSG/GSH	*p* Value	*R* ^2^
Handgrip strength	+2.3 (+0.51, +4.5)	**0.04**	0.30
ASM/height^2^	+1.9 (−0.6, +4.3)	0.12	0.02
Gait speed	+0.9 (−1.8, +3.6)	0.49	0.00
Neutrophils	+1.5 (+0.2, +2.8)	0.07	0.01
NLR	+4.8 (+1.2, +6.8)	**0.002**	0.42
IL-1α	+2.1 (+0.1, +4.3)	0.06	0.03
IL-6	+2.9 (+0.6, +5.2)	**0.02**	0.15
TNF	+1.3 (+0.2, +2.7)	0.09	0.01
VEGF	+0.5 (−2.7, +1.8)	0.39	0.00
MCP-1	+0.3 (−0.2, +2.8)	0.48	0.00

Abbreviations: SD, standard deviation; GSSG, oxidized glutathione; GSH, reduced glutathione; ASM, appendicular skeletal muscle mass; NLR, neutrophil-to-lymphocyte ratio; IL, interleukin; TNF, tumor necrosis factor; VEGF, vascular endothelial growth factor; MCP-1, monocyte chemoattractant protein-1.

## Data Availability

The study data are available on reasonable request to the corresponding authors. The data are not publicly available due to ethical and privacy restrictions.

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
