# Peer review of "Sarcopenia Is Associated with Changes in Circulating Markers of Antioxidant/Oxidant Balance and Innate Immune Response"

_antioxidants, 2023, doi:10.3390/antiox12111992_

Round 1

Reviewer 1 Report

Comments and Suggestions for Authors

I am grateful for the opportunity to review the paper entitled "Sarcopenia is associated with changes in circulating markers of antioxidant/oxidant balance and innate immune response ".

The authors have done a great work to define and relate modifications in circulating markers of redox homeostasis and innate immune response in human sarcopenia, which makes it of special interest due to the scarcity of literature on the subject.

This is a cross-sectional study, with a sample of 72 subjects (32 sarcopenic), which is an important limitation, as the authors have indicated, and one that greatly limits the scope of the study, since the study parameters can be affected by multiple factors not directly related to sarcopenia.

Authors have made a great effort to attempt to hypothesize about the physiological mechanisms that may justify their results, based on some previous studies.

Therefore, and in order to improve the manuscript, I believe that the authors should pay attention to these aspects:

1.       Add a sample size calculation that justifies the statistical power of the results obtained. If not, they should calculate the effect size of the differences that justify the results obtained.

2.       If the sample is not representative, why didn't they extend the dates for patient collection?

3.       Justify the high incidence of sarcopenia in the study population, since the figures in Europe are considerably lower than those present in this study. Has a previous screening or selection of patients been made? If so, justify why and how such pre-selection may influence the results.

4.       The walking speed is calculated, according to EWGSOP2, in a 4m corridor, not 6m.

5.       Finally, given that the main finding of the study is the alteration of the circulating redox balance markers, I would suggest that the authors add a predictive model that would allow us to know which blood and functional-clinical variables would allow us to predict their behavior, in order to improve the detection of oxidative stress and, with this, to make an early prevention of the clinical condition through these markers.

I thank you again for the opportunity to review this work.

Author Response

I am grateful for the opportunity to review the paper entitled "Sarcopenia is associated with changes in circulating markers of antioxidant/oxidant balance and innate immune response ".

The authors have done a great work to define and relate modifications in circulating markers of redox homeostasis and innate immune response in human sarcopenia, which makes it of special interest due to the scarcity of literature on the subject.

This is a cross-sectional study, with a sample of 72 subjects (32 sarcopenic), which is an important limitation, as the authors have indicated, and one that greatly limits the scope of the study, since the study parameters can be affected by multiple factors not directly related to sarcopenia.

Authors have made a great effort to attempt to hypothesize about the physiological mechanisms that may justify their results, based on some previous studies.

Therefore, and in order to improve the manuscript, I believe that the authors should pay attention to these aspects:

  1. Add a sample size calculation that justifies the statistical power of the results obtained. If not, they should calculate the effect size of the differences that justify the results obtained.

Reply: we thank the reviewer for his positive comments. We calculated the effect size of the differences at 90% power and 95% level of significance, and a sample size of 28 subjects was required. This calculation is now included in the revised manuscript (lines 166-168).

  1. If the sample is not representative, why didn't they extend the dates for patient collection?

Reply: the sample resulted as representative, so that we did not need to extend patient collection.

  1. Justify the high incidence of sarcopenia in the study population, since the figures in Europe are considerably lower than those present in this study. Has a previous screening or selection of patients been made? If so, justify why and how such pre-selection may influence the results.

Reply: we observed a higher frequency of sarcopenia in our sample population because the enrolment was performed in a hospital department (rather than overall population). We added this point as a limitation of our study in the discussion section (lines 418-421).

  1. The walking speed is calculated, according to EWGSOP2, in a 4m corridor, not 6m.

Reply: corrected as suggested (lines 108-109).

  1. Finally, given that the main finding of the study is the alteration of the circulating redox balance markers, I would suggest that the authors add a predictive model that would allow us to know which blood and functional-clinical variables would allow us to predict their behavior, in order to improve the detection of oxidative stress and, with this, to make an early prevention of the clinical condition through these markers.

I thank you again for the opportunity to review this work.

Reply: we thank the reviewer for his suggestion. Accordingly, we performed a series of crude linear regression models to predict association between variables of muscle function/mass/performance or immune response markers with GSSG/GSH (chosen as redox balance marker). The main text of the manuscript was modified consequently (lines 179-188, lines 313-320, Table 6 and its legend).

Reviewer 2 Report

Comments and Suggestions for Authors

In the current study, the authors evaluated the role of the circulating markers of oxidants and antioxidants and innate immune response in the development of sarcopenia. The authors report impaired redox balance in sarcopenic patients and higher neutrophil-to-lymphocyte ratio, interleukin (IL)-4, IL-6, IL-10, and tumor necrosis factor (TNF) when compared with the non-sarcopenic subjects.

Questions/ suggestions/ limitations of the study.

Introduction:

It is known that age-related sarcopenia is associated with skeletal muscle denervation. Denervation leads to increased oxidative stress in muscle. None of this was mentioned in the Introduction or the Discussion.

Methods:

Line 106: “who could independently perform walking speed, grip strength, and muscle mass” test?

Results:

Table 1. There is a significant difference in insulin concentration within the Sarcopenia group (from 4 to 36). Could you please provide information whether there was a correlation between high insulin and decreased muscle performance, increased oxidative stress and pro-inflammatory markers.

Discussion

Line 299: “Furthermore, our data showed decreased expression of SOD1 and CAT which encode for two main endogenous antioxidant enzymes. Overall, these reports are consistent with reduced systemic antioxidant defense as possible cause of redox disbalance in sarcopenia”.  These measurements were done in the circulating mononucleated cells. Sarcopenia is age-related loss of muscle mass. Caution should be taken when interpreting these data. Mybe increased total-body oxidative stress and decreased antioxidants are causing sarcopenia instead of sarcopenia causing oxidative stress? No direct measurements of oxidative stress in muscle were done in this study.

Line 309:  No direct measurements of pro-inflammatory markers were done in muscle samples. The study showed that there is a correlation between sarcopenia and circulating in blood pro-inflammatory markers. What is primary: the sarcopenia is causing an increase in the pro-inflammatory markers, or the pro-inflammatory markers are causing sarcopenia?

The authors should clearly describe in the discussion that they report the correlation of sarcopenia with the increased oxidative stress and pro-inflammatory markers and decreased antioxidants. The direct relationship between them is unclear. Maybe the significantly increased insulin levels in patients with sarcopenia is the reason for the increased oxidative stress and pro-inflammatory markers?

Sarcopenia and age-related muscle denervation and its relationship to the increased oxidative stress and pro-inflammatory markers should be discussed.

Comments on the Quality of English Language

Minor editing of English language required.

Author Response

In the current study, the authors evaluated the role of the circulating markers of oxidants and antioxidants and innate immune response in the development of sarcopenia. The authors report impaired redox balance in sarcopenic patients and higher neutrophil-to-lymphocyte ratio, interleukin (IL)-4, IL-6, IL-10, and tumor necrosis factor (TNF) when compared with the non-sarcopenic subjects.

Questions/ suggestions/ limitations of the study.

Introduction:

It is known that age-related sarcopenia is associated with skeletal muscle denervation. Denervation leads to increased oxidative stress in muscle. None of this was mentioned in the Introduction or the Discussion.

Reply: we thank the reviewer for his valuable comments. We mentioned denervation-related oxidative stress in the introduction of revised manuscript (lines 56-58).

Methods:

Line 106: “who could independently perform walking speed, grip strength, and muscle mass” test?

Reply: the sentence was modified (lines 113-115).

Results:

Table 1. There is a significant difference in insulin concentration within the Sarcopenia group (from 4 to 36). Could you please provide information whether there was a correlation between high insulin and decreased muscle performance, increased oxidative stress and pro-inflammatory markers.

Reply: to comply with the reviewer’s request, we performed correlation tests between insulinemia and muscle performance, GSH, GSSG/GSH, HNE- and MDA-protein adducts, CRP, NLR and pro-inflammatory cytokines, but no significant results were reported. Results are shown in the revised Figure 2 and described in the following text (line 288, lines 298-301).

Discussion

Line 299: “Furthermore, our data showed decreased expression of SOD1 and CAT which encode for two main endogenous antioxidant enzymes. Overall, these reports are consistent with reduced systemic antioxidant defense as possible cause of redox disbalance in sarcopenia”.  These measurements were done in the circulating mononucleated cells. Sarcopenia is age-related loss of muscle mass. Caution should be taken when interpreting these data. Mybe increased total-body oxidative stress and decreased antioxidants are causing sarcopenia instead of sarcopenia causing oxidative stress? No direct measurements of oxidative stress in muscle were done in this study.

Line 309:  No direct measurements of pro-inflammatory markers were done in muscle samples. The study showed that there is a correlation between sarcopenia and circulating in blood pro-inflammatory markers. What is primary: the sarcopenia is causing an increase in the pro-inflammatory markers, or the pro-inflammatory markers are causing sarcopenia?

Reply: to comply with the reviewer’s observation, we added a statement in the discussion about cautious interpretation of our results, since we did not directly measure redox balance in skeletal muscle (lines 368-370).

The authors should clearly describe in the discussion that they report the correlation of sarcopenia with the increased oxidative stress and pro-inflammatory markers and decreased antioxidants. The direct relationship between them is unclear. Maybe the significantly increased insulin levels in patients with sarcopenia is the reason for the increased oxidative stress and pro-inflammatory markers? 

Reply: we stated in the discussion that we could not establish a clear causal relationship between oxidative stress and alteration of immune response (lines 404-406). Since we did not find any significant correlation between insulin levels and markers of oxidative stress/inflammation, we avoided discussing this point.

Sarcopenia and age-related muscle denervation and its relationship to the increased oxidative stress and pro-inflammatory markers should be discussed.

Reply: we included this point in the discussion (lines 335-338).

Reviewer 3 Report

Comments and Suggestions for Authors

Ballanti and co-workers show that sarcopenia in humans is associated with various measures of redox status and inflammation. Given the prevalence of sarcopenia, it is important to better understand its aetiology and pathogenesis. However, while the current study brings some new insights, it is not clearly delineated form authors’ own previous work, which focused on the same issues.

General comments (major):

1.      Results of the current study should be put in a clearer context of their own previous work (Maturitas, 2018). The novelties in the current study as opposed to the previous study should be highlighted. This is really important because the previous study looked into similar parameters, including GSH, GSSH, TNF, HNE and MDA adducts, GSSG/GSH ratio, as well as a range of other parameters and tests that were examined also in the current study. Authors cite their own previous work twice in the current manuscript, but this seems insufficient given the relevance of the previous publication with regard to the finding in the current one.

2.      The authors have examined a range of laboratory parameters and performed correlation analysis of these parameters (Figure 2). They also assessed different aspects of muscle mass and function. Unfortunately, they did not link the two sets of data. It would be much more important to show whether laboratory parameters, such as concentrations of GSH, GSSH, interleukins, correlate with muscle mass and function or physical performance.

3.      Conclusions should be firmly based on the results. For instance, interpretations, such as “Results from the present study confirm our previous observation and strengthen the role of redox homeostasis in maintenance of muscle structure and function.”, would be much stronger if the authors demonstrated that markers of redox status correlated with those of muscle mass and function.

4.      To our knowledge, this is the first investigation demonstrating that alterations of circulating markers of redox balance are associated with modifications of immune response in human sarcopenia.” Authors should refer to and discuss previously published work, such as studies on GSH and/or GSSH levels in plasma or muscle.

5.      Graphs should indicate individual measurements in addition to means and SE.

Specific comments:

1.       Differences in measures related to muscle strength, quantity, and performance, as well as results related to multidimensional evaluation, are reported in Table S1. Sarcopenic patients exhibited lower handgrip strength, ASM/height2, and gait speed with respect to people with no sarcopenia.” Since this is a paper on sarcopenia, measures of muscle function should be reported in the Results rather than in a supplementary table.

2.      Of note, scores from all the tests included in our geriatric multidimensional evaluation protocol were reduced in sarcopenia as compared to no sarcopenia group.” Which tests? Results of relevant tests should be shown.  Alternatively, the reference to these tests should be omitted.

3.      Despite being used before, diagnostic criteria for (severe) sarcopenia which were used in this study should be clearly reported in the Methods.

4.      How was physical activity assessed and how was low physical activity defined?

5.      The unit of ESR (mm/s) is incorrect.

6.      CRP unit (mg/dL) is probably wrong. Did authors mean mg/L? 21.5 mg/dL would indicate highly elevated CRP levels.

7.      Unit “n/mm3” should be explained. Does n mean number? Usually this would be written as “/mm3 or mm-3”.

8.      Which form of vitamin D was measured (vitamin D or 25-(OH)-vitamin D)?

9.      The concentrations of vitamin D were rather low. Could this explain or contribute to muscle weakness in the sarcopenia group?

10.   Were patients with sarcopenia insulin-resistant? Since both glucose and insulin were measured authors could calculate indices of insulin resistance, such as HOMA.

11.    To verify this hypothesis, the analysis of genes encoding for enzymes related to GSH synthesis/recycling (GR and GS), as well as other antioxidant enzymes (SOD1 and CAT)…” It would be better for the readers if authors explained what the abbreviations mean when they introduced here. Also, the authors may wish to consider using a scheme showing how enzymes contribute to metabolism of GSH.

12.   The expression of GS, GR, SOD1, and CAT was lower in PBMCs of subjects with sarcopenia. However, the concentrations of leucocytes were different between the two groups. Could this explain the different expression levels? Again, although PBMC is a standard abbreviation it would be better to explain it when first using it in the Results (otherwise readers need to search for it in the Methods).

13.   What was the concentration of monocytes?

14.   α2-globulins are reported in Table 3, suggesting that authors performed electrophoresis of serum proteins. If the did, why not include information regarding other fractions, especially the immunoglobulins as well.

15.   Are VEGF and EGF markers of immune response as stated in the Results?

16.   skeletal muscle is the greatest oxygen utilizer in human body” This statement should be corrected or clarified. Muscles use approximately 20 % of oxygen, which is more or less matched by the brain, while the digestive system uses more than 20 %.

17.   Please correct: “neutrophil-to-lymphocyte ration”.

Author Response

Bellanti and co-workers show that sarcopenia in humans is associated with various measures of redox status and inflammation. Given the prevalence of sarcopenia, it is important to better understand its aetiology and pathogenesis. However, while the current study brings some new insights, it is not clearly delineated form authors’ own previous work, which focused on the same issues.

General comments (major):

  1. Results of the current study should be put in a clearer context of their own previous work (Maturitas, 2018). The novelties in the current study as opposed to the previous study should be highlighted. This is really important because the previous study looked into similar parameters, including GSH, GSSH, TNF, HNE and MDA adducts, GSSG/GSH ratio, as well as a range of other parameters and tests that were examined also in the current study. Authors cite their own previous work twice in the current manuscript, but this seems insufficient given the relevance of the previous publication with regard to the finding in the current one.

Reply: we thank the reviewer for his comment. In our previous study, we merely reported changes in circulating redox markers occurring in sarcopenic patients. We expanded our observation by studying the relationship between such changes and alterations of circulating markers of inflammation and innate immunity. To comply with the reviewer’s observation, 2e emphasized the novelty of this study in the introduction and discussion section of the revised manuscript (lines 53-54, line 65, line 341, lines 407-408).

  1. The authors have examined a range of laboratory parameters and performed correlation analysis of these parameters (Figure 2). They also assessed different aspects of muscle mass and function. Unfortunately, they did not link the two sets of data. It would be much more important to show whether laboratory parameters, such as concentrations of GSH, GSSH, interleukins, correlate with muscle mass and function or physical performance.

Reply: to comply with the reviewer’s request, we performed correlation tests between indexes of muscle function/mass/performance and circulating markers of redox balance as well as inflammation/immunity, which are represented in the revised version of Figure 2 and commented in the following text (lines 298-301).

  1. Conclusions should be firmly based on the results. For instance, interpretations, such as “Results from the present study confirm our previous observation and strengthen the role of redox homeostasis in maintenance of muscle structure and function.”, would be much stronger if the authors demonstrated that markers of redox status correlated with those of muscle mass and function.

Reply: we agree with the reviewer’s comment. We found such correlation (please see reply to point 2).

  1. “To our knowledge, this is the first investigation demonstrating that alterations of circulating markers of redox balance are associated with modifications of immune response in human sarcopenia.” Authors should refer to and discuss previously published work, such as studies on GSH and/or GSSH levels in plasma or muscle.

Reply: to comply with the reviewer’s request, we referred to studies that evaluated GSH and GSSG levels in plasma and muscle in the revised discussion (lines 352-355).

  1. Graphs should indicate individual measurements in addition to means and SE.

Reply: we modified all the graphs in Figure 1, showing also raw data measurements, as requested.

Specific comments:

  1. “Differences in measures related to muscle strength, quantity, and performance, as well as results related to multidimensional evaluation, are reported in Table S1. Sarcopenic patients exhibited lower handgrip strength, ASM/height2, and gait speed with respect to people with no sarcopenia.” Since this is a paper on sarcopenia, measures of muscle function should be reported in the Results rather than in a supplementary table.
  2. “Of note, scores from all the tests included in our geriatric multidimensional evaluation protocol were reduced in sarcopenia as compared to no sarcopenia group.” Which tests? Results of relevant tests should be shown.  Alternatively, the reference to these tests should be omitted.

Reply: we removed Table S1 from the Supplementary material and included its results in Table 2 of the revised manuscript (lines 215-223).

  1. Despite being used before, diagnostic criteria for (severe) sarcopenia which were used in this study should be clearly reported in the Methods.

Reply: to clearly report diagnostic criteria of sarcopenia, we extended their description in the Methods, as suggested (lines 102-105, lines 108-111).

  1. How was physical activity assessed and how was low physical activity defined?

Reply: physical activity was assessed by self report activity logs, as cited in the Methods of the revised manuscript (lines 85-86).

  1. The unit of ESR (mm/s) is incorrect.
  2. CRP unit (mg/dL) is probably wrong. Did authors mean mg/L? 21.5 mg/dL would indicate highly elevated CRP levels.
  3. Unit “n/mm3” should be explained. Does n mean number? Usually this would be written as “/mm3or mm-3”.

Reply: we corrected the ESR unit (mm/h) and CRP unit (mg/L) in table 4, and we explained that “n” means “number” in the legend (line 262).

  1. Which form of vitamin D was measured (vitamin D or 25-(OH)-vitamin D)?
  2. The concentrations of vitamin D were rather low. Could this explain or contribute to muscle weakness in the sarcopenia group?

Reply: we measured 25-(OH)-vitamin D; table 1 and the main text were corrected (line 121, line 207). We cannot argue about the role of 25-(OH)-vitamin D in sarcopenia, because its low levels could either contribute to muscle weakness or be the consequence of sarcopenia-associated altered lifestyle (reduced exposure to sunlight, inappropriate dietary regimen…). In our modest opinion, discussion of this point goes beyond the scope of our manuscript. 

  1. Were patients with sarcopenia insulin-resistant? Since both glucose and insulin were measured authors could calculate indices of insulin resistance, such as HOMA.

Reply: we calculated HOMA-IR index, as suggested by the reviewer, and modified Table 1 and the main text accordingly (lines 123-124, line 208).

  1. “To verify this hypothesis, the analysis of genes encoding for enzymes related to GSH synthesis/recycling (GR and GS), as well as other antioxidant enzymes (SOD1 and CAT)…” It would be better for the readers if authors explained what the abbreviations mean when they introduced here. Also, the authors may wish to consider using a scheme showing how enzymes contribute to metabolism of GSH.

Reply: we explained abbreviations and inserted a scheme, as suggested (lines 244-246, Figure S1)

  1. The expression of GS, GR, SOD1, and CAT was lower in PBMCs of subjects with sarcopenia. However, the concentrations of leucocytes were different between the two groups. Could this explain the different expression levels? Again, although PBMC is a standard abbreviation it would be better to explain it when first using it in the Results (otherwise readers need to search for it in the Methods).
  2. What was the concentration of monocytes?

Reply: even though gene expression patterns could be influenced by quantitative variations of PBMC subsets, we did not find significant differences of lymphocyte and monocyte count between the two groups. We explained PBMC abbreviation in the text (lines 247-248) and reported the concentration of monocytes in Table 4 of the revised manuscript.

  1. α2-globulins are reported in Table 3, suggesting that authors performed electrophoresis of serum proteins. If the did, why not include information regarding other fractions, especially the immunoglobulins as well.

Reply: we included information regarding immunoglobulins in Table 4 of the revised manuscript

  1. Are VEGF and EGF markers of immune response as stated in the Results?

Reply: VEGF and EGF, more than being growth factors and regulators of angiogenesis, are often used as surrogate markers of immunity in clinical studies (https://doi.org/10.1111/bju.12068, https://doi.org/10.3390/ijms24054922,  https://doi.org/10.7554/eLife.44478, https://doi.org/10.1161/CIRCRESAHA.116.309279)

  1. skeletal muscle is the greatest oxygen utilizer in human body” This statement should be corrected or clarified. Muscles use approximately 20 % of oxygen, which is more or less matched by the brain, while the digestive system uses more than 20 %.

Reply: in resting conditions, muscles consume almost 30% of body’s total volume of oxygen consumption, according to the following table:

From “Clark JM, Lambertsen CJ. Pulmonary O2 toxicity: a review. Pharmacol Rev 1971;23(2):37-133.”

  1. Please correct: “neutrophil-to-lymphocyte ration”.

Reply: corrected.

Round 2

Reviewer 1 Report

Comments and Suggestions for Authors

I thank the authors for the effort made in the revision of this document, for having tried to incorporate the suggestions for improvement.

I have 2 relevant doubts:

The calculation of the sample size does not seem adequate, since the authors only indicate the following: A sample size of 28 subjects 166 in each group was calculated to detect the difference at 90% power and 95% level of significance. They should detail the significant difference in which variable and what it consists of based on previous studies.

After the regression analysis, with the sample limitation, and due to the low predictive value of the variables, which means that the model has a very low R2, the question arises as to whether the statement that "correlation does not imply causation" should be applied.

Author Response

I thank the authors for the effort made in the revision of this document, for having tried to incorporate the suggestions for improvement.

I have 2 relevant doubts:

The calculation of the sample size does not seem adequate, since the authors only indicate the following: A sample size of 28 subjects 166 in each group was calculated to detect the difference at 90% power and 95% level of significance. They should detail the significant difference in which variable and what it consists of based on previous studies.

Reply: according to our previous work (Bellanti F et al., Maturitas, 2018), we reported a mean difference in GSSG/GSH of 10.35. This is now detailed in the methods section (lines 167-168).

After the regression analysis, with the sample limitation, and due to the low predictive value of the variables, which means that the model has a very low R2, the question arises as to whether the statement that "correlation does not imply causation" should be applied.

Reply: we agree with the reviewer about the limitations of our study. We discuss our data by cautiously referring to a “relationship”, but we are not able to establish a cause-effect association (lines 403-406). Our observation encourages advanced mechanistic research to define whether reactive species trigger disruption of innate immune response, or whether age-related altered innate immunity leads to redox disbalance in sarcopenia (lines 411-414).

Reviewer 2 Report

Comments and Suggestions for Authors

The authors addressed all of my comments.

Author Response

The authors addressed all of my comments.

Reply: we thank again the reviewer for his comments, which helped improving the quality of our manuscript

Reviewer 3 Report

Comments and Suggestions for Authors

Bellanti and co-workers have thoroughly revised the manuscript. They have considered all my suggestions and provided a satisfactory explanation or revised the manuscript accordingly. The addition of new results, especially HOMA and correlations between laboratory and clinical parameters, as well as more transparent data presentation, are especially appreciated. The supplementary figure also nicely supports the manuscript. I think all revisions have significantly benefited the manuscript.

My only remaining comment concerns the oxygen/energy usage in skeletal muscle. Unfortunately, I cannot find the Figure that they pasted in their rebuttal in the Clark & Lambertsen (1971) paper (is this the correct reference?).

If authors wish to highlight the energy usage in rodents, I agree with their statement that skeletal muscles use the greatest fraction of energy in the body. However, since this is a human study, I thought that human data might be more relevant.

With regard to the estimation of energy/oxygen usage in humans vs. rodents, authors may wish to consult a somewhat more recent paper than Clark & Lambertsen (1971) – i.e. the review by Rolfe and Brown (Cellular Energy Utilization and Molecular Origin of Standard Metabolic Rate in Mammals, Physiol Rev, 1997), especially Table 1 on p. 735, which shows that both brain and muscle use 20% of oxygen, while the liver and the gastrointestinal tract (combined) use 27%.

Finally, I think the statement regarding the fraction of oxygen usage requires a better reference than the current ref. 36.

Author Response

Bellanti and co-workers have thoroughly revised the manuscript. They have considered all my suggestions and provided a satisfactory explanation or revised the manuscript accordingly. The addition of new results, especially HOMA and correlations between laboratory and clinical parameters, as well as more transparent data presentation, are especially appreciated. The supplementary figure also nicely supports the manuscript. I think all revisions have significantly benefited the manuscript.

My only remaining comment concerns the oxygen/energy usage in skeletal muscle. Unfortunately, I cannot find the Figure that they pasted in their rebuttal in the Clark & Lambertsen (1971) paper (is this the correct reference?).

If authors wish to highlight the energy usage in rodents, I agree with their statement that skeletal muscles use the greatest fraction of energy in the body. However, since this is a human study, I thought that human data might be more relevant.

With regard to the estimation of energy/oxygen usage in humans vs. rodents, authors may wish to consult a somewhat more recent paper than Clark & Lambertsen (1971) – i.e. the review by Rolfe and Brown (Cellular Energy Utilization and Molecular Origin of Standard Metabolic Rate in Mammals, Physiol Rev, 1997), especially Table 1 on p. 735, which shows that both brain and muscle use 20% of oxygen, while the liver and the gastrointestinal tract (combined) use 27%.

Finally, I think the statement regarding the fraction of oxygen usage requires a better reference than the current ref. 36.

Reply: we are enormously grateful to the reviewer for his suggestions. We consulted the indicated review, and we changed the statement related to the fraction of oxygen usage (line 361) as well as reference 36 in the revised manuscript.